# Using Blockchain to Ensure Trust between Donor Agencies and NGOs in Under-Developed Countries

**Ehsan Rehman** [1], **Muhammad Asghar Khan** [1], **Tariq Rahim Soomro** [1,*], **Nasser Taleb** [2], **Mohammad A. Afifi** [3] **and Taher M. Ghazal** [3,4]

1. College of Computer Science and Information Systems, Institute of Business Management, Karachi 75190, Pakistan; ehsan.rehman@iobm.edu.pk (E.R.); muhammad.asghar@iobm.edu.pk (M.A.K.)
2. Faculty of Management, Canadian University of Dubai, Dubai 117781, United Arab Emirates; nasser.taleb@cud.ac.ae
3. School of IT, Skyline University College, Sharjah 1797, United Arab Emirates; mohammed.afifi@skylineuniversity.ac.ae (M.A.A.); taher.ghazal@skylineuniversity.ac.ae (T.M.G.)
4. Center for Cyber Security, Faculty of Information Science and Technology, Universiti Kebansaan Malaysia, Bangi 43600, Malaysia
* Correspondence: tariq.soomro@iobm.edu.pk

**Abstract:** Non-governmental organizations (NGOs) in under-developed countries are receiving funds from donor agencies for various purposes, including relief from natural disasters and other emergencies, promoting education, women empowerment, economic development, and many more. Some donor agencies have lost their trust in NGOs in under-developed countries, as some NGOs have been involved in the misuse of funds. This is evident from irregularities in the records. For instance, in education funds, on some occasions, the same student has appeared in the records of multiple NGOs as a beneficiary, when in fact, a maximum of one NGO could be paying for a particular beneficiary. Therefore, the number of actual beneficiaries would be smaller than the number of claimed beneficiaries. This research proposes a blockchain-based solution to ensure trust between donor agencies from all over the world, and NGOs in under-developed countries. The list of National IDs along with other keys would be available publicly on a blockchain. The distributed software would ensure that the same set of keys are not entered twice in this blockchain, preventing the problem highlighted above. The details of the fund provided to the student would also be available on the blockchain and would be encrypted and digitally signed by the NGOs. In the case that a record inserted into this blockchain is discovered to be fake, this research provides a way to cancel that record. A cancellation record is inserted, only if it is digitally signed by the relevant donor agency.

**Keywords:** blockchain; ensuring trust; NGOs; encryption

## 1. Introduction

Blockchain is not a new technology. Digital signatures were introduced in 1991 to secure the integrity of the documents which are mostly considered the foundation of blockchain [1]. Satoshi Nakamoto proposed one of the important applications of blockchain by introducing Bitcoins in 2008 [2]. Within the last few years, after the exploration of the benefits of blockchain, governments and industries have been using it in various domains such as supply chain, identity management, recordkeeping [3], and education [4].

Blockchain is a decentralized technology that allows different stakeholders to access and replicate a database. The database can only be updated using predetermined rules, and once changed, it is shared with all parties. Each transaction in the blockchain is connected in a chain to make sure that everybody has the most up-to-date version of the ledger. On a blockchain network, a distributed ledger is a method for replicating and storing transactional data. The distributed ledger expands on this idea by replicating data across several nodes [5].

Blockchain removes the need for third-party providers to verify transactions because it is a peer-to-peer network that timestamps them. This type of recordkeeping is tamper-resistant as each peer has a copy of the complete ledger and new transactions can only be added with the consensus of the majority of peers or following the predetermined rules [3].

A typical blockchain consists of three parts: block, chain, and network. The block contains transactions that store information about some important activity, such as tracking goods or assets, sensitive medical information, or critical information generated by machines using IoT [6]. When a new blockchain network is formed then few rules are defined. These rules administer the working of the network and set the details, such as the size of the transaction in each block, the addition of transactions, etc. When the agreed number of transactions fills a block then that block is chained with the previous blocks using a hash value. A hash is a one-way algorithm that generates a fixed value that is unique for the transactions in that block, and the same hash can only be generated if the block contains the same transactions. The hash value would be different if a block of data is altered by someone during the transmission or modified by any peer. A different hash value shows that the data in the original block have been modified and data are not trustworthy anymore. Multiple hash values can be combined and hashed together to form a single hash or Merkle root. A Merkle tree is created by adding more hashes to the base. A simplified version of the blockchain is illustrated in Figure 1 below [3].

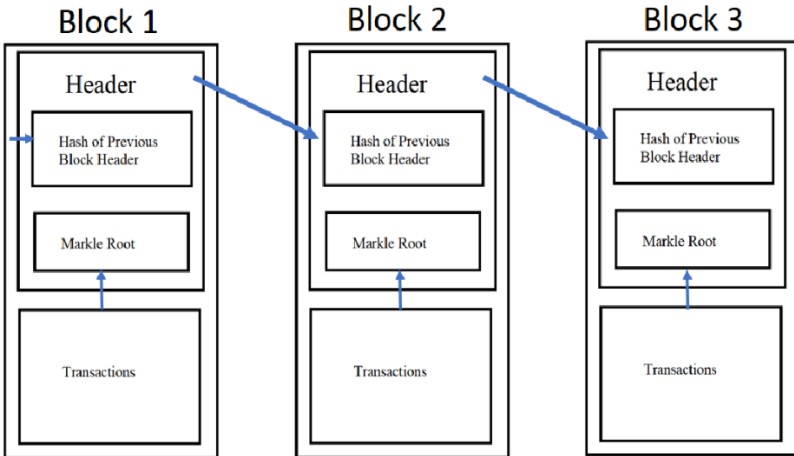

**Figure 1.** Simplified Blockchain [3].

Blockchain platforms allow developers to develop blockchain applications. Many blockchain platforms with different features are available. The few common blockchain platforms are Bitcoin, Ethereum, Hyperledger, R3, Ripple, and Electro-Optical System (EOS) [7]. The selection of blockchain platform is highly dependent on various factors such as industry focus (financial services, digital asset management, cross-industry), ledger type (permissioned, permissionless), consensus algorithm (proof of work, pluggable framework, probabilistic voting, majority voting, chain-based Byzantine fault tolerant, Stellar consensus protocol), support of smart contracts and type of governance who managed the network. Bitcoin is the most famous blockchain platform. Bitcoin assumes that there is no trust between the parties, it facilitates a large number of decentralized nodes to ensure that the blockchain is not tampered with by cybercriminals. Miners, who are active users, manage the decentralized nodes. Miners are needed by cryptocurrency platforms to solve crypto puzzles as proof of work, which is then validated by other miners or nodes. Miners who solve and verify the puzzles are paid in cryptocurrency. Although miners are needed for cryptocurrency platforms, they are not required for other blockchain platforms [8].

Joe and Raafat [9] ranked the usage of blockchain in different domains based on recent peer-reviewed publications. The top ten areas of blockchain application based on their

popularity in descending order are IoT, energy, health care, finance, resource management, government, exchange, transportation, BPM, and right management.

There is widespread consensus that education is critical for improving livelihoods and economic prosperity in developed countries [10]. Many developing countries are dependent on local and international donors due to scarcity of resources and other governance issues. Organizations, such as Association for Childhood Education International (ACEI), Education International, Save the Children, UNESCO, UNICEF, etc., work across continents and barriers to ensure that every child receives a high-quality education. Some donor agencies have lost their trust in the local NGOs in under-developed countries, as some NGOs have been involved in the misuse of funds. This is evident from irregularities in the records. For instance, in education funds, on some occasions, the same student has appeared in the records of multiple NGOs as a beneficiary, when in fact, a maximum of one NGO could be paying for a particular beneficiary. Therefore, the number of actual beneficiaries would be smaller than the number of claimed beneficiaries as the same recipients are enjoying the benefits from the same/multiple NGO/s at the same time. This research proposes a blockchain-based solution to ensure that no beneficiary will receive the same benefit multiple times by the same NGO or by another NGO. The following sections provide a summary of the related work; the next section will explore research material and method of the proposed model; the next section discusses the structure and Maintenance of the Ledger, and finally paper concludes.

## 2. Related Work

One of the promising areas of blockchain is the record-keeping of charity organizations' activities. A lot of attempts [8,11–21] have been made to find a way for a proper charitable platform to run based on blockchain technology. Muhammad, Misbah and Adnan [12] claimed that the current systems are unstructured, vague, and lacking in donor confidence. In the case of donations made to various individuals by NGOs, there is no adequate record keeping, and the presence of certain dishonest individuals within the charity organizations has caused donors to lose faith and trust in this social cause [11]. The donors have no idea whether their contributions are being used properly or not. Other factors that cause donors to lose faith in charity are negligence and mismanagement. Many religions, such as Christianity and Islam, believe in mandatory religious donations to underprivileged masses. A recent study [12] highlighted religious philanthropy scandals of neglect and mismanagement of donations recorded in the United States, the United Kingdom, the Kingdom of Saudi Arabia, and Pakistan, among other places, making it a global concern.

Smart contracts are the main feature of blockchain 2.0 that can help to automate many aspects of supply chain management and contract execution using technologies such as Etherium and IOTA (open-source distributed ledger). Many humanitarian organizations, such as UNICEF and the World Food Program, are looking into the possibilities of cash-based assistance and digital identification [11]. Every year, hundreds of millions of dollars are lost to corruption, and blockchain is now being used to monitor aid funding to stop the flow of black money [11]. Ching-Sheng Hsu et al. [13] proposed an E-Voucher System for Supporting Social Welfare Using blockchain Technology. In their model, they discussed the major problems associated with paper vouchers, which are used by donors to avoid paying cash, and suggested that vouchers are digitized and blockchain technology be used to overcome the flaws of paper vouchers.

Jia Hongwei and Deng Xiuquan [14] analyzed the characteristics and application of blockchain in detail and offered a model to solve problems in the domain of social emergency relief. Rizal Mohd Nor et al. [15] proposed a blockchain-based donation system to solve the issues of third-party extra fees, payment processing delays, and accountability. Adalberto Rangone and Luca Busolli [16] examined the idea of Charity Wall [17] and suggested that different aspects of blockchain technology can be used to strengthen the philanthropic system. Furthermore, Wang 'Jia and Chen Haifeng studied the blockchain-based charity application in China and acknowledged that many common issues related to

donation systems can be avoided by using blockchain. Baokun Hu and He Li [18] proposed another charity system that uses Ethereum as a blockchain platform.

## 3. Material and Method

NGOs are given funds by international donor organizations to deal with various issues in developing countries, such as education, medicine and health, earthquake relief, and more. Due to the lack of checks and balances in developing countries, the purpose for which the funds are being utilized can be misreported. There are various ways in which this can occur. The most common possibility is that of claiming that a certain amount of funds is required for a given project when significantly lesser funds are required for it.

Unless the project is well-defined, and hence the amount of funds required is within a reasonably small range, it is hard to detect whether the amount asked is reasonable, overquoted, or underquoted. This research, therefore, focuses on only well-defined projects, where the requirement of funds can be reasonably bounded. To be specific, the example of educational funds would be used for illustrative purposes in this article, however, the methodology outlined applies to any well-defined project.

### 3.1. Overview of the Proposed Solution

Trust between NGOs and donor agencies is managed in a distributed system with restricted access. Using consensus [22,23], only those entities are allowed to join the network that can be verified as valid NGOs or donor agencies. The verification includes looking up the public keys of known NGOs and donor agencies and confirming that the given entity is one of these valid entities, using digital signatures.

The transactions between the donor agencies and the NGOs are recorded in a distributed ledger. For this article, transactions are defined as funds that are promised/committed by a donor agency to an NGO. The distributed software system that is managing the ledger, allows only those transactions to be inserted that fulfill certain properties. It is these properties that have to be defined carefully, to ensure that incorrect, fraudulent, and/or duplicate records are not inserted.

What properties should a transaction have before it is allowed into the ledger? Firstly, each transaction is digitally signed both by the NGO that is claiming that a sum of money is required, and by the donor agency that is providing the funds. Secondly, the purpose of the funds should be specified, and the number of funds quoted should fall within the allowed range for the given purpose. Thirdly, the transaction should not be a duplicate transaction. For example, an NGO cannot claim to be funding the lodging for a student X, in year Y (using unique ID), when another NGO earlier has claimed to be doing the same. Lastly, the node generating and proposing the transaction should have recently contributed to the network by staying connected to the network, and being an active member. This criterion is explained in detail, later in this article.

The above makes it clear that there is a table that is required, that stores a list of all possible "purposes" for which a fund can be allocated. For each such "purpose", the table also includes a range. If an amount is used for a given purpose $p$, then this amount must fall within the prescribed range for $p$. For instance, one cannot claim that a million dollars are going to be used for helping a student buy textbooks throughout his or her undergraduate degree. This table, therefore, ensures that the monetary amount and the purpose, both of which are part of an inserted transaction, are consistent with each other before insertion is allowed.

This paper differentiates between the two words "transaction" and "record". Whereas a transaction is a monetary agreement between an NGO and donor agency, along with other requisite details, a record is similar to a row of data in a classical database. In the case of the ledger, the record could include a transaction (in which case we call it a "transaction record"), or it could be invalidating a previous transaction (in which case we call it an "invalidation record"). The reason that differentiating between "transaction" and "record"

is required is that transaction records are not the only type of records that are inserted into the ledger.

It should be possible to insert "invalidation" records, whose purpose is solely to invalidate an earlier record in the ledger. Readers should note that deletion from a ledger is not an allowed operation, therefore an additional "invalidation" record is inserted. The reason for such invalidation is that an inaccuracy in the inserted transaction could have subsequently come to light. Hence, a relevant authority, such as the donor agency that promised to provide those funds, can insert a digitally signed invalidation record.

### 3.2. Joining and Leaving the Distributed System

The distributed system is made up of two kinds of nodes: the NGOs and the donor agencies. It is important to make the distinction between these two types, as the purpose of these two types of nodes is different from each other (Figure 2). For instance, an NGO is not allowed to be listed as a donor agency in a transaction. Additionally, it allows certain restrictions to be imposed, such as: only a donor agency (and not an NGO), is allowed to insert an invalidation record.

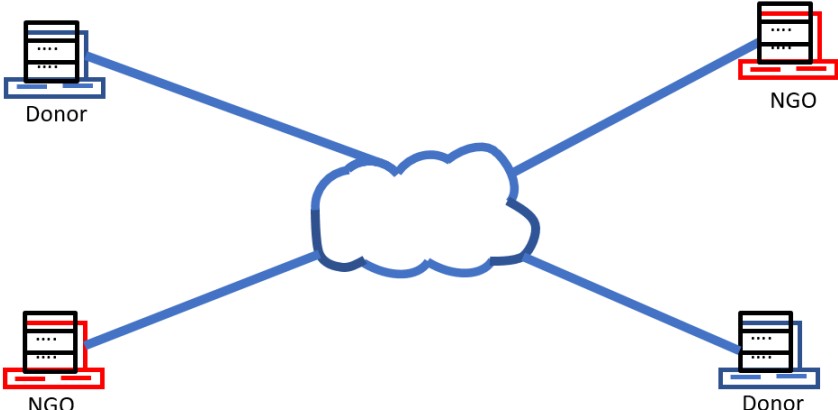

**Figure 2.** Distinguishing two kinds of nodes in the network: NGOs and donors.

The nodes in the system participate in the maintenance of the ledger and some other consensus algorithms. For consensus algorithms to be reliable, two things need to be ensured. Firstly, that the votes are cast only by valid NGOs and donor agencies, and secondly that the number of votes is large enough.

To ensure secrecy of the data in the ledger, as well as to ensure that the votes are cast only by NGOs or donor agencies, the network needs to be restrictive as to what nodes can become part of the network. The system cannot allow every node to join and cast votes. If everyone was allowed into the network, then because the number of NGOs and donor agencies is much smaller than the total number of computing nodes in the world, hence the consensus vote could easily be hijacked by groups of nodes that are neither NGOs nor donor agencies, as they will constitute the majority. In short, the consensus vote would become open to attack.

The system also needs to ensure that the number of voting nodes is not too small, which would make the consensus unreliable. For example, if there are just two nodes that are currently in the network, then they can team up and make any unwarranted changes to the ledger, such as remove the previous block of record.

### 3.3. Restricting Who Can Join the Network

For simplicity, let us assume that at time t = 0, the network starts with a number of nodes that is not too small. Additionally, that the joined nodes are valid NGOs and donor agencies. Now suppose a new NGO node n wants to join. A new node n can join the network as a donor agency, or as an NGO. In both cases, n needs to be allowed into the network via a distributed consensus algorithm.

The new node n has to prove that it is a valid NGO (and/or donor agency) so that the current members can allow it to join the network. The requesting node n sends a request to one of the members already in the network (step 1 in Figure 3a). This member then broadcasts the request to the rest of the members (step 2 in Figure 3b). For this to be possible, each of the current members must have a list of known NGOs and/or donor agencies. This list is not centralized, nor is it a part of the distributed ledger, but is maintained separately by each of the nodes, and therefore could be different for each of the nodes. Hence, one node may know about a given NGO (and/or donor agency) m, but the other node does not know about m. However, if the majority of the NGO (and/or donor agency) members connected to the network, know about m, then m can be allowed to join the network. In Figure 3b, the new member has been rejected by a majority vote of 3:1.

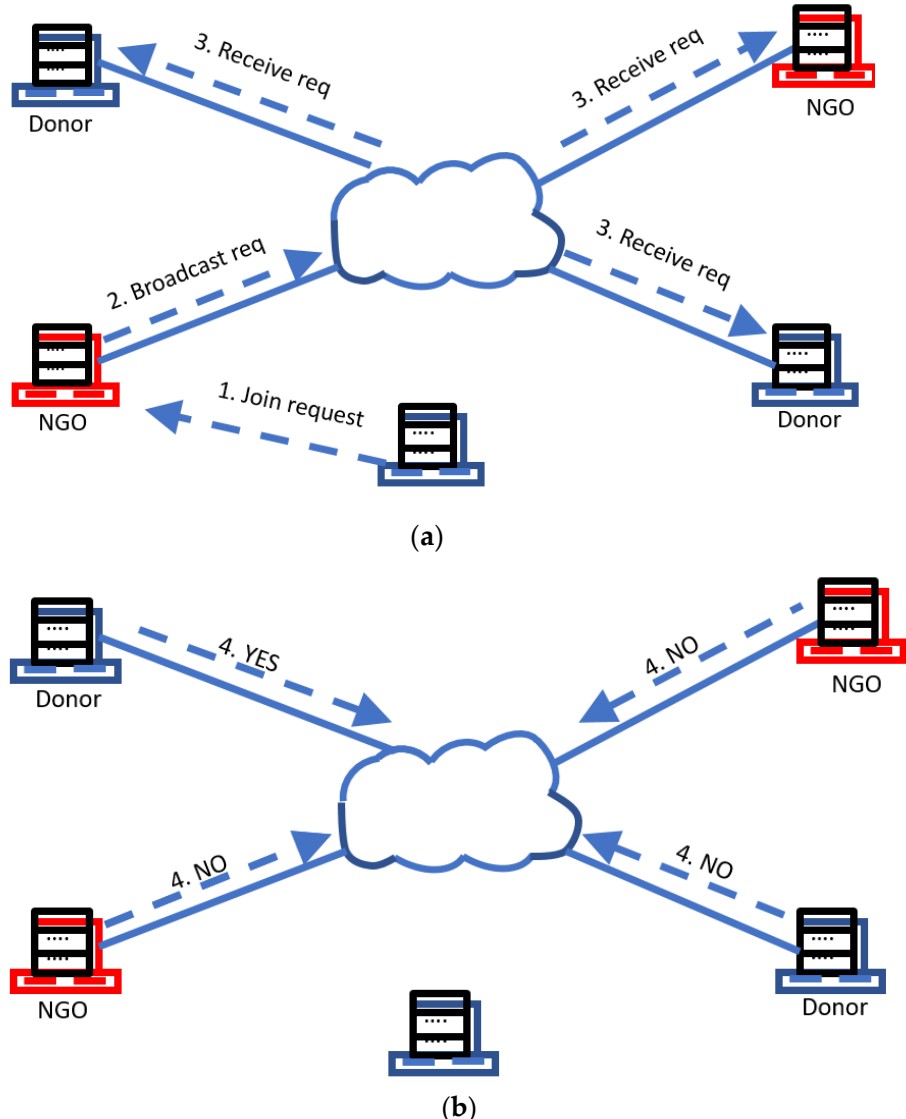

**Figure 3.** (**a**) A new node wants to join as a donor. The request is received by all members. (**b**) The new member is rejected with a 3:1 vote.

If the number of members currently in the network is too small, then the voting mechanism outlined above, is not carried out. In such a case, the new joining node should be a well-known NGO or donor, whose identity should be available on a reliable centralized list. Otherwise, the new node is not accepted into the network.

The new node desiring to join the network, declares its identity through a public key or a certificate. Certificates rely on external certification authorities, although there has been recent work on developing consensus with only public keys and digital signatures, and without requiring any certificates [24].

### 3.4. Algorithm for Joining the Network

Algorithm 1 shows the procedure for accepting a new node into the distributed network of nodes. The nodes in this network are either NGOs or donors. Suppose a node *c* wants to join the network. The protocol starts with *c* sending a message to any node *b* already connected to the network.

When node *c* sends a join request to node *b*, the function b.sendJoin(node c) is called at node *b*. The function name is prepended with "*b*" to show that the function is being executed locally at node *b.*

The node *b* starts a voting procedure that will decide whether *c* will be allowed to join or not. In order to perform this, *b* first updates the list of known nodes in the network (lines 10 and 11). This list $L_b$, stored locally at *b*, is the list of all nodes currently in the network that are known to *b*. Any of the standard node discovery algorithms can be used to populate this list.

If the number of donors in $L_b$, or if the number of NGOs in $L_b$ is too small (line 12), then there are not enough voters for the result of voting to be reliable. In such a case, the new node *c* is allowed to join the network only if it is a well-known NGO or donor, which can be verified from some known centralized servers (line 13–15). If c has been successfully added to $L_b$, then this event is broadcasted by *b* to all the other nodes in $L_b$, so that the rest of the nodes will also add the node *c* into their respective lists (after possibly verifying *c* from central servers).

---

**Algorithm 1** To allow a node to join the distributed network

---

1　　b.computeDecision(join, c):

2　　　　broadcast(computeSelfVote(join, c))

3　　　　**For each** j ∈ list $L_b$ of known nodes in the network

4　　　　　receive(j, vote, join, c) //*has a timeout*

5　　　　countTotalVotes(join, c)

6　　　　**If** total NGO votes > v1 **and** total donor votes > v2

7　　　　　insert c into $L_b$

8　　　　**End**

9　　b.receiveJoin(node c):

10　　　**If** the list of online nodes, $L_b$, is outdated

11　　　　update $L_b$

12　　　　**If** numbNGO < n1 **or** numbDonors < n2

13　　　　　**if**(isWellKnown(c))

14　　　　　　add c to $L_b$

　　　　　　　broadcast("c has joined the network")

15　　　**Else**

16　　　　broadcast(joinRequest, c)

17　　　　computeDecision(join, c)

18　　　**End**

---

In the case that the number of nodes in $L_b$ is large enough, then the join request of $c$ is broadcasted to other nodes in $L_b$ (line 16). Each of these nodes is then going to call the function computeDecision(join, c).

The function computeDecision(join, c), when executed at a node $b$, decides whether $c$ is added to $L_b$ or not. The result depends on voting. In line 2, the node $b$ first computes its own vote of whether $c$ should join the network or not. This local vote depends on whether $b$ knows $c$ to be a trustable NGO or donor, or not. This local vote is broadcasted to the other known nodes in the network (line 2). In lines 3–4, the votes of the other known members are collected. Note that the receive on line 4 has a timeout. That is, there is a maximum time period for which $b$ waits for these messages.

In lines 5–8, the node $b$ counts the number of "yes" votes. If these votes cross a pre-decided threshold value, then the join request is accepted, and $c$ is added to $L_b$. Note that such a list $L_f$ exists at every node $f$. The other nodes that received enough votes, will also add $c$ to their respective lists. Hence, these nodes will contact $c$ for votes and consensus in the future.

Each node has a list L stored locally, and these lists can be exchanged among the nodes. Hence, those nodes that could not add a node to their lists, can do so later. A node $f$ will add $c$ to its list $L_f$, if $c$ is already in the lists of several nodes that are contained of $L_f$, and if $f$ can successfully communicate with $c$ over the network.

### 3.5. Encouraging the Nodes Not to Leave the Network

For the majority vote in consensus algorithms to be reliable, the majority should be made of a sizable number of votes. In other words, the number of nodes in the network should not be too small. Therefore, what is needed is a mechanism to encourage the relevant NGOs and donor agencies to remain connected, and participate as active members in this distributed system.

The main idea to encourage the nodes to stay connected is a penalty mechanism, whose details are in the subsection below. However, the essence of this mechanism can be summarized as follows: if an entity (NGO or donor agency) has not been online for a long time, and suddenly decides to come online and insert a record, then its record is not inserted into the ledger until the entity has spent its due time connected and participating in the network.

### 3.6. A Penalty Mechanism to Encourage Nodes to Stay Connected

This research proposes a penalty mechanism to encourage the nodes to stay online and connected to the network. As the network is reasonably small, made only of known NGOs and donor agencies, it can be assumed that each current member in the network knows the identity of all the other members that are currently connected.

Let $t_n$ be the number of hours over the last d days, for which the node n was connected to the distributed network of NGOs and donors. Each node $b$, approximates the value of $t_n$ for every node n in the network. If there are N + 1 nodes, then each node $b$, stores N tables: $T_{b1}, T_{b2}, \ldots T_{bN}$. Each table $T_{bi}$ has $24 \times d$ binary entries, each entry represents an hour over the last $d$ days. An entry of $T_{bi}$ is "1", if according to the knowledge of $b$, the node $i$ was connected to the network during that hour. Hence, summing all the entries in $T_{bi}$ gives the total number of hours that $i$ was connected to the network (according to the belief of $b$).

The two rows shown in Table 1 above, show two such tables, say, $T_{bi}$ and $T_{bj}$. Suppose that the entity with public key "FF2G", which is shown in the second row of the table above, broadcasts a new record $r$, so that it can be inserted into the next block of the ledger. Denote the entity with public key "FF2G", as node $c$. Even if $r$ is a valid record from node $c$, the node $b$ would not include $t$ in the next block that $b$ is mining. This is because $c$ might have not spent sufficient time connected to the network. To decide whether to insert $r$ into the next block, the node $b$ will look into its table (shown above), and total the number of hours during which $b$ could observe $c$ as being online.

**Table 1.** Table of online connectivity.

| Public Key | Hour 1 | Hour 2 | Hour 3 | Hour 24 × d |
|:---:|:---:|:---:|:---:|:---:|
| ABA14S | 0 | 1 | 0 | 1 |
| FF2G | 1 | 0 | 1 | 1 |

This scheme would encourage the nodes to stay connected, and contribute. If the node $c$ has not spent enough time on the network, then all properly functioning nodes will not include the record $r$ in the blocks they are mining. There are two ways in which the record $r$ can be inserted into a block. (i) The node $c$ spends enough time on the network, and then re-broadcasts the record $r$. (ii) The node $c$ adds $r$ to the block it is mining and wins the race to add the next block to the blockchain. Note that in (ii), the node $c$ has to carry out useful work for the blockchain before $r$ is added. while in (i) the node $c$ has a penalty of needing to wait before $r$ is accepted into the blockchain.

The algorithm, as well as the probabilistic analysis of the algorithm, is described in detail later in this article, in Section 4.2.2.

## 4. The Structure and Maintenance of the Ledger

The information in the ledger is transparent, meaning that any computing node connected to the distributed system can view the contents. As the nodes are the NGOs and donor agencies, it implies that the NGOs and donor agencies can see each other's data that are placed in the ledger.

It can also be allowed for the data not to be as transparent as mentioned above. For instance, it might not be desirable for one NGO to know the transaction details of the other entities. Everything detailed in this article can be adapted easily to such a scenario. The details are given in Section 4.4.

### 4.1. The Different Kinds of a Record in the Ledger

In the real world, a transaction between an NGO and a donor agency can also be canceled. The reason could be an inaccuracy or fraudulency in the transaction that only later comes to light. For example, the donor agency might realize later, that the student for which the funds were being used, is not going to the given university anymore. It could also be the case that the transaction details were accurate, however, the student expired. This, along with many other possibilities, implies that the system should allow the cancellation of transactions.

The above makes it clear that the distributed ledger needs to have at least two kinds of records: transaction records, and transaction-cancellation records. The details of the information in each type of record are given in the subsections to follow.

#### 4.1.1. The Transaction Record

There are at least four pieces of information that define a transaction between an NGO and a donor agency: The NGO, the donor agency, the amount of money in the fund, and the purpose of the fund (see the transaction T in Figure 4a). The two variables, the amount of money in the fund, and the purpose are intrinsically linked. A large sum of money cannot be allocated for a purpose that can be accomplished in far less an amount. Similarly, the amount should also not be much smaller than what would be needed for a given purpose. This link between purposes and amounts is provided by the "Table of Purposes" which is described in Section 4.4. Note that this table of purposes is also responsible for preventing duplicated transactions; two transactions in the ledger cannot have identical purposes. For instance, it should not be allowed for two different transactions to be paying the same student, for exactly the same purpose.

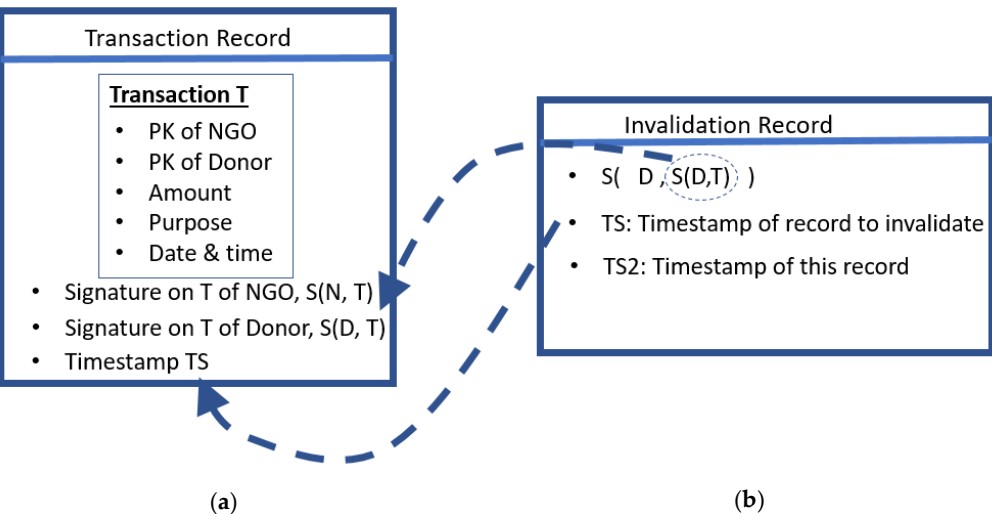

**Figure 4.** (**a**) The transaction record T, (**b**) Invalidation record of T. The arrows show the references to previous data.

Other than the basic transaction information, the transaction record should also include digital signatures of the NGO and the donor agencies involved (see Figure 4a). Note that a transaction may involve more than one donor agency. Although the purpose of a transaction is singular, there could be funds from various agencies to fulfill the purpose. In such a case, the transaction needs to list the amounts against each of the donor agencies, and the digital signatures of all the involved parties need to be included in the record.

It is not possible for a party that is not involved in the transaction to add its signature on the transaction, as that would make the transaction record invalid. This is because the transaction includes the list of all involved parties. A digital signature S on the transaction implies that S agrees with all the details of the transaction, including the list of all parties. A transaction record is valid, only when the set of parties that have signed the transaction, is the same as the set of parties listed inside the transaction. Any difference between these sets would lead to the transaction being rejected. See Figure 4a, where the signatures of the NGO and donor is on the data block T, and these data themselves contains the list of NGO and donor. If T was altered in any way, it would invalidate the signature of the original participating NGO and donor, as their signatures are on T. Hence, a party not involved in the transaction cannot add its signature without invalidating the record.

### 4.1.2. The Invalidation Record

If a donor agency wants to invalidate a previous transaction (possibly fake), it needs to declare that it is pulling out of it, by inserting an invalidation record. Let us say that a donor agency D, wants to invalidate a transaction in record R. To have authority to invalidate R, agency A must be one of the original signatories in R.

The invalidation record includes only the signature of the donor agency D, on its signature of R. Let S(X, Y), be the sign of entity X, on data Y. Hence, S(D, T) is the signature of the entity D, on the transaction T in record R. To invalidate its signature on record R, the entity D would insert a record with the following signature: S(D, S(D, T)) (See Figure 4b). This signature, along with the timestamp of the record that is being invalidated (TS in Figure 4a,b), is the invalidation record. The reason for having the timestamps is explained in the next subsection. In the case that multiple donor agencies were involved in the transaction, the pulling out of a donor agency D only invalidates the corresponding part of the deal.

Invalidation of a transaction T by a signatory should be signaled to the other signatories of R. Such an invalidation should alert the other donor agency (and NGO), that something could be wrong with the transaction, and that they should also consider pulling out of it. As the invalidation record is inserted by consensus of the nodes in the network, a

certain t number of these nodes can be selected to send a message to the other signatories of R, that one of the members is pulling out.

The selection of these t members can be achieved by consensus. Each member broadcasts a message m, as well as the acknowledgment of each such message m received from each of the members. Next, using Lamport's clock [25], the messages m can be ordered [26,27]. That is, every node will agree on the same order of the messages. Now, the senders of the first t of these messages, will take responsibility and inform the other signatories about the invalidation of the transaction R.

The reason for selecting t members, rather than just 1 member, is to achieve a degree of reliability under adversarial attack.

The timestamp on a record (TS in Figure 4a, and TS2 in Figure 4b), is the time at which its insertion was proposed by some node in the network. The records inserted into the ledger should have timestamps, firstly because it is important for history and diagnosis. Secondly, timestamps help in faster processing. Before inserting an invalidation record that is invalidating a previous transaction T, the existence of T should be verified. Adding timestamps would help in finding T much faster. These timestamps would be added according to the consensus of the nodes in the network, at the time of insertion of the record into the ledger.

The participating nodes in the network would have downloaded the ledger, which by default, is sorted according to timestamps. Therefore, it would be possible to apply search algorithms such as binary search, to the data. When a proposed invalidation record refers to an earlier signature, it also includes the timestamp of the record that is being invalidated (In Figure 4b, TS refers to the timestamp of the transaction record).

As the network has a much smaller number of nodes, as compared to bitcoin, timestamps can be added by consensus. To add consensus-based timestamps, one could implement the strategy in [28]. A simpler strategy can also be implemented where every node broadcasts the proposed timestamp. Now that every node has all the proposed timestamps from the other nodes, the median value of the timestamps can be taken as the consensus value of the timestamp. This assumes, however, that there are no misbehaving nodes, that are sending a different value to different nodes. Hence, each node would compute a different median. In such a case, a leader can be elected using a leader-election algorithm [27,29,30]. The median calculated by the leader node is then taken to be the timestamp, by a majority vote: a node n agrees to the suggested value of the leader if it is not too far from the median calculated by n.

## 4.2. A Single Block of Transaction in a Ledger

### 4.2.1. Addition of a Block into the Ledger

Blocks can be mined by any node (NGO or donor), connected to the network. Similar to bitcoin, the node has to solve a sufficiently hard mathematical problem, before this block is accepted into the blockchain by the other members of the network.

Similar to bitcoin, several records are first collected into blocks, and then these blocks are inserted into the blockchain. Each block has a timestamp of the time at which it was inserted into the ledger.

New records are proposed by nodes in the network, and just as in bitcoin, by distributed consensus, a subset of these records is selected to become part of the next block added to the ledger. If a proposed record $r$ is invalid, then a properly functioning node $b$, will not add it $r$ to the new block that $b$ is processing. Hence, if the next block in the blockchain is added by $b$, this block will not contain $r$. If a block contains an invalid record $r$, then the nodes receiving this record will ignore this block, and not add the block to its local blockchain.

### 4.2.2. Rejection of Invalid Records

There are four main reasons for which a transaction or invalidation record is considered invalid.

(a) Inconsistent or duplicated data: Firstly, the data in the transaction could be inconsistent or duplicated. For instance, maybe there are three parties mentioned in the transaction, but only two of them have signed it. Another possibility is that the amount of money involved in the transaction is not consistent with the purpose mentioned for the transaction.

(b) Unknown signatories: If at least one of the signing parties is not a known NGO or donor agency, then the record cannot be inserted into the ledger. Note that there can be disagreement among the nodes whether a public key belongs to a known NGO/donor agency, or not. One node might know about X, while another might not. Hence, an agreement will be reached among the nodes, by voting.

(c) Insufficient online time: Algorithmically, the most interesting possibility is that although the record is correct, none of the signatories listed in the record have spent enough time and effort in the network to be allowed to insert their transactions. Recall, that to be allowed to insert a record, the entity, which is either an NGO or donor agency, must have spent enough time in the network, verifying the new blocks of data, as well as being part of other consensus algorithms.

(d) Invalid reference to a previous record: In the case of an invalidation record, there should be a reference to a previously inserted record, along with its timestamp. If such a record does not exist, then the invalidation record cannot be inserted into the ledger.

Each new record can be given a sequence number, with an agreement among the nodes on the sequence number of a given record. This numbering is not difficult, because there is an agreement among the nodes on the order of arrival of the new proposed records.

The nodes need to decide, which of the records are suspicious and should not be inserted into the block. As the network is going to be reasonably small, with a manageable number of nodes, the following solution is possible: there is a consensus using voting, for each record to be inserted into the next block.

Another option, which is less communication-intensive, is to perform consensus on only those records that look suspicious and hence should be rejected. One of the nodes multicasts the sequence number of the transaction that the node believes should be rejected. Subsequently, all the nodes communicate with each other and use voting to decide whether this particular transaction is accepted or rejected into the next block.

Inconsistent or Duplicate Data

The data of a transaction can be inconsistent for the following reasons:

(a) Incorrect signatures: Either there are some parties listed in the transaction that have not signed the transaction, and/or one of the signatures is not from a party listed inside the transaction. In short, the following two sets are not identical: the set of parties mentioned in the transaction details and the set of parties from which the signatures have been received.

(b) Purpose and amount mismatch: Each transaction has to say the purpose that the NGO would use this money for. If the quoted amount is either too small or too large and therefore not in the allowed range of the given purpose, then there is a mismatch in the purpose and amount.

(c) Duplicated purpose: If a transaction that has been added to the ledger, allocated funds for the same purpose, and to the same entity (such as the same student), then it is a duplicate/fake transaction and would be rejected. For instance, two different transactions could not be allocating funds to buy the same books, for the student with the same ID during the same year of his/her education. The details of detecting duplicate transactions are given in Section 4.4.

(d) Purpose unknown: The purpose of an amount can only be one of the allowed purposes. The allowed purposes are listed in the "Table of purposes" that will be covered in Section 4.4 in this article. Each purpose in this table has a unique identifier, as well as a description. A transaction lists its purpose using one of these unique identifiers,

and also includes the description of the given purpose. If this pair of (unique identifier, description), does not exist in the table of purposes, then the transaction record cannot be accepted into the ledger.

(e) One of the required fields is missing: Lastly, if one of the above-required fields is missing, then the record should not be inserted into the ledger.

Insufficient Online Time

An earlier section described how to penalize nodes that have not spent time on the network, by not accepting their proposed records into the new blocks. This section describes the procedure in more detail, along with probabilistic analysis.

Every node $b$ approximates the total duration (over the last $d$ days) for which every other node $c$ was connected to the distributed network. Every node $b$, perform this for all nodes c that are known to $b$.

Every node $b$ stores a $24 \times d$ binary table, Tbc, for every other node $c$. If $b$ knows about $n$ other nodes in the network, then there are $n$ such tables stored at $b$. Each cell in this $24 \times d$ table represents an hour over the last $d$ days. The node $b$ messages $c$ periodically during every hour. If $c$ acknowledges the majority of the messages from $b$, then $b$ marks $c$ as "1" for that hour, otherwise, the value is "0". In the case that $b$ itself is not connected during the hour, $b$ cannot mark any node with a "1" for that hour.

Functions in Algorithms 2 and 3 are invoked at node $b$, when $b$ receives a record $r$ from $c$. The record $r$ can potentially to be inserted into the next block in the blockchain. If $r$ is invalid, then it will be ignored. However, if $r$ is valid, then its insertion depends on whether $c$ has spent enough time connected to the distributed network.

---

**Algorithm 2** updating the online time of node c

```
        b.computeTime(node c):
1           table = getTable(c)
2           sum = 0
3           for i from 1 to d   //taking data of the last d days
4               for j from 1 to 24
5                   sum = sum + table[i][j]
6           return sum
7           end

        b.updateTime(node c):
8           table = getTable(c)
9           for j ∈ S, where S ⊆ Lb
10              otherTable = receiveTable(j, c)
11              for i from 1 to d   //taking data of the last d days
12                  for j from 1 to 24
13                      if otherTable[i][j] == 1
14                          table[i][j] = 1
15          end
```

---

The function Compute Time(c), in Algorithm 2 above, calculates the total number of hours the node $c$ has been online, according to the table $T_{bc}$ stored at $b$. Line 1 in Algorithm 2, retrieves this table, and stores it in the variable "table". Lines 2–5, totals the number of cells in this table where the value is 1. This sum approximates the total number of hours that $c$ was connected to the network, over the last $d$ days.

A node $b$ should not just rely on its local table $T_{bc}$ to calculate the duration $c$ was connected for. One reason is that $b$ itself might not have been connected for some of those times. Another possibility is that messages were lost between $b$ and $c$, making $b$ think that $c$ is offline.

In Algorithm 2, the function updateTime(node c), is a function that is used by a node $b$ to update its local table $T_{bc}$. In summary, this procedure for updating is as follows: for each of the $24 \times d$ hours in $T_{bc}$, if any other node saw $c$ online during that hour, then $b$ also marks $c$ to be online during that hour. This is carried out by writing a "1" to the appropriate cell of the table $T_{bc}$.

In line 8, the local table $T_{bc}$ is fetched and stored in the variable "table". In lines 9–14, $b$ contacts a subset S of the nodes in the list $L_b$. For each node $j \in$ S, the table $T_{jc}$ stored at $j$ is fetched, and stored in the local variable "otherTable" (line 9). If any cell value of $T_{jc}$ is "1" (line 12), then the corresponding cell value $T_{bc}$ is also set to 1 (line 13).

Note that in a large system, nodes might know a lot of other nodes. If each node $b$ starts exchanging its tables with all the other nodes, it can flood the system with too many messages. Therefore, this strategy is not scalable to a large network. However, the network of NGOs and donors is not going to be too large. Additionally, in the case of a large network, each node b can be restricted to communicate with only $k$ other nodes in order to update the table $T_{bc}$.

---

**Algorithm 3** Accepting or rejecting a record r from c

|   | b.isAccepted(node c, record r): |
|---|---|
| 1 | updateTime(c) |
| 2 | t = computeTime(c) |
| 3 | **if** t > threshold |
| 4 | │ accept(r) |
| 5 | **else** |
| 6 | │ reject(r) |
| 7 | **end** |

---

Algorithm 3 uses the functions earlier introduced in Algorithm 2, to accept or reject a record $r$ from a node $c$. The function isAccepted(c, r) is invoked at node $b$, when $b$ receives a record $r$ from $c$. In line 1, the locally stored table $T_{bc}$ is updated using the function updateTime(c). Next, on line 2, the total online time of $y$ is calculated using this updated table. Note that on line 2, the function updateTimes(c) is called, whose details have been given earlier in Algorithm 2. On line 2, the variable "$t$" stores the total number of hours that $c$ was seen connected to the network. If $t$ is above a given threshold (line 3), then the record $r$ is accepted (line 4). By "accepted" it is meant that $b$ will insert record $r$ in the next block (of the blockchain) mined by $b$. Otherwise, the record is "rejected" by $b$ (line 6).

Let us now analyze: (i) whether there is a possibility of wrongfully accepting a record $r$ even when the proposing node $c$ has not spent enough time in the network, and (ii) whether there is a possibility of wrongfully rejecting a record $r$ even when $c$ has indeed spent enough time in the network.

**Case 1**: The node $c$ proposing record $r$, has not spent enough time on network.

With Algorithm 3, if the time spent in the network by a node $c$, $t_c \leq$ threshold, then no other properly functioning node is going to accept the record $r$ proposed by $c$. This is because only those entries of a table $T_{bc}$ can be "1" if $c$ was observed online by some node during the corresponding hour. As $c$ has not been online for enough number of hours, therefore not enough entries in $T_{bc}$ would be "1". Therefore, $b$ would not accept $r$. That is, $b$ would not insert $r$ into the next block that $b$ is mining.

When an otherwise valid record $r$ is rejected by $b$, because the proposing node had not spent enough time on the network, the only impact is that a block that is mined by $b$,

would not include $r$. If some other node mines a new block and includes $r$ in that block, $b$ would consider it as a legal block and add it to its local blockchain.

In such a case, there are two ways in which $r$ will be inserted. Firstly, the node $c$ can stay connected to the network, until $t_c$ > threshold. Once $t_c$ > threshold, the node $c$ can broadcast $r$ once again. This time, it is more likely to be added to the block by some node. The second possibility is that $c$ adds a block to the blockchain. In that case, $c$ can add $r$ to the block. This is fair, as $c$ had to perform useful work for the blockchain, before adding the record $r$.

**Case 2**: Even if a node has spent enough time, its record can be wrongfully rejected.

The actual number of hours that $c$ was connected to the network, $t_c$, will usually be underestimated when approximated by another node $b$. The node $b$ counts the number of hours during which it could communicate with $c$. However, $b$ itself might not have been online for several of those hours to observe $c$. To alleviate this problem, $b$ also consults $|S|$ other nodes, but those $|S|$ nodes are also underestimating the value of $t_c$.

It is therefore possible that a node $c$ has spent enough time, yet its proposed record $r$ is not included in the next block. The analysis below calculates the probability of this undesirable event.

Calculating the probability of this undesirable event.

Let the "threshold" value, in Algorithm 3, be denoted by $v$. In other words, if $t_c$ > $v$, then the record $r$ should be accepted, as $c$ has spent enough time in the network. Let $f(j)$ = $t_{jc}$, be the (approximation) of $t_c$ that has been computed at node $j$. Note that $f(j)$ can be any integer from 0 to $t_c$ (inclusive). In the analysis below, it is assumed that for every node $j$, $f(j)$ is uniformly distributed between 0 and $t_c$. This is a pessimistic assumption as $f(j)$ is more likely going to be closer to the true value $t_c$ than to 0, unless $j$ is usually offline.

Note that in Algorithm 3, a node $b$ is performing the following steps: $b$ communicates with $|S|$ other nodes, and obtains a sample, $(t_{1c}, t_{2c}, t_{3c} \ldots t_{Nc})$. The sample has size N = $|S|$. Each value in the sample is an approximation of the value of $t_c$ at nodes 1, 2, … N, where $t_c$ is the time for which $c$ was connected to the network. Using the notation in the above paragraph, this sample can be written as: $(f(1), f(2), \ldots f(N))$. Next, the value of $t_{bc}$ is set to a value, at least as large as max($t_{1c}, t_{2c}, t_{3c} \ldots t_{Nc}$) = max($f(1), f(2), \ldots f(N)$).

As each $f(j) \sim$ unif(0, $t_c$), therefore $t_{bc}$ is a maximum of N such random variables. In other words, $t_{bc}$ is the Nth order statistic, in a sample of N identically distributed, uniform random variables. Assuming that these variables are independent, it becomes a sample of N, independent and identically distributed (IID) random variables, each having a unif(0, $t_c$) distribution.

**Theorem 1.** *Let the value of $t_c$ = $v$ + $k$. In other words, c has spent k more hours than required by the threshold value v. Let each $f(j) \sim$ unif(0, $t_c$) = unif(0, v + k). Suppose b obtains a sample of N such random variables. Let probability that b rejects the record r proposed by c be denoted by Pr(b rejects r). Then:*

$$\Pr(b \text{ rejects } r) \leq \left( \frac{v}{v+k} \right)^{\text{N}} \tag{1}$$

**Proof.** As $b$ obtains a sample of size N, the value of $t_{bc}$ is at least as large as M = max($f(1), f(2), \ldots f(N)$). Therefore, if M > $v$, then $b$ is going to accept $r$. That is, Pr($b$ rejects $r$) $\leq$ Pr(M $\leq$ $v$).

As M is the maximum of N IID uniform random variables, the probability distribution of M can be computed. Let $g(x)$ be the probability density function of the maximum value in a sample of N IID random variables, each having a PDF of $f(x)$, and CDF of $F(x)$ respectively.

It is common knowledge that: $g(x) = Nf(x)F(x)^{\text{N}-1}$. As in the current case, $f(x)$ and $F(x)$ are the PDF and CDF of unif(0, $v$ + $k$), we obtain: $f(x) = \frac{1}{v+k}$, and $F(x) = \frac{x}{v+k}$. Substituting these above, we obtain: $g(x) = N\frac{1}{v+k} \left( \frac{x}{v+k} \right)^{\text{N}-1}$.

The corresponding CDF is found by integration:

$$G(x) = \int_0^x g(t)dt = \mathrm{N}\frac{1}{(v+k)^{\mathrm{N}}}\int_0^x t^{\mathrm{N}-1}dt = \frac{x^{\mathrm{N}}}{(v+k)^{\mathrm{N}}} \qquad (2)$$

This therefore, is the CDF of M. The probability that $\mathrm{M} \le v$, is hence given by: $G(v) = \left(\frac{v}{v+k}\right)^{\mathrm{N}}$.

This proves: $\Pr(b \text{ rejects } r) \le \left(\frac{v}{v+k}\right)^{\mathrm{N}}. \square$

This is an encouraging result, as the probability reduces exponentially with N, the number of nodes that are sampled. Let us take some reasonable values. Suppose $v = 100$ h. That is, a node $c$ has to be online for 100 h (say, over a period of 10 days), before its record $r$ can be accepted. Suppose the node $c$ was online for 105 h, which is only 5 h above the threshold value. Hence, $k = 5$. Suppose N = 30; that is $b$ contacts 30 other nodes. Then the probability that $b$ (wrongfully) rejects $r$, is $\le \left(\frac{100}{105}\right)^{30} = 0.23$. That is, there is a 77% chance that the very next block will contain $r$.

Lastly, note that the above theorem only gives the probability of a single node rejecting a record $r$. The connected nodes can perform a majority vote on whether to accept record $r$ or not. In that case, the probability of rejection is the same as the probability, that more than $T/2$ nodes reject $r$, where T is the total number of nodes in the network. This probability becomes vanishingly small, and is of the order of

$\Pr(r \text{ is rejected by consensus}) = \left(\left(\frac{v}{v+k}\right)^{\mathrm{N}}\right)^{T/2} = \left(\frac{v}{v+k}\right)^{\mathrm{N}\times T/2}.$

### 4.2.3. The Frequency of Adding New Blocks

Transaction and invalidation records are inserted into the ledger in groups or blocks, just as in bitcoin. These blocks are chained together, and this chain is called the blockchain. However, there are some questions to be answered, such as how many records should be there in a single block? How often should the blocks be inserted, and how is the decision reached on which records will be included in the next block?

NGOs and donor agencies should not have to wait an undue time before their transaction appears on the ledger, although the urgency is not as high as in some applications of bitcoin payments. For instance, payment with bitcoin at a coffee shop requires quick confirmation of the transaction. However, the level of urgency is lesser, in the case of confirming a transaction between NGOs and donor agencies.

The next block is added to the ledger if any of the following two conditions are true:

(1)  The waiting time for some outstanding transaction or invalidation has crossed the maximum allowed value.
(2)  The number of outstanding transactions has crossed the maximum allowed value.

The first point above ensures that no valid transaction has to wait an undue amount of time before being inserted into the ledger. The second point ensures that the block does not become unduly large when being inserted. If the block size is very large, then it will stress the network communication to process such a large block. As an example, in bitcoin, the block sizes are kept under 1 Mb of size.

Note that the threshold of time in point (a) above, is selected to be large enough so that the new transactions can be processed with a typical, average processor. This is preferred so that the nodes in the network can verify the transactions and agree to include the block of records into the ledger.

Once a node notices that at least one of the above two conditions is satisfied, it waits for a small random amount of time, and then sends the list of the sequence number of the outstanding records that should be included into the next block. This list is the "recommendation" from this node for the contents of the next block. This list is only sent if

it did not receive any such list proposed from another node as yet. The reason for waiting for a random amount of time is to avoid flooding the network by all the nodes sending their proposed lists at the same time. If multiple valid lists are indeed received, then using Lamport's logical clocks, these lists can be ordered among the nodes, and the first one in this ordered sequence of lists is selected to be added as the next block.

### 4.3. Table of Purposes and Detecting Duplicates

A transaction cannot be considered valid unless the amount quoted is within a valid range for the given purpose. The table of purposes lists the valid range of amounts for a given purpose. The following are the basic fields of a row in the table of purposes:

(1)    The unique ID of the purpose: This could just be the row number.
(2)    The name of the purpose: This serves the purpose as an English title of the purpose that people can understand.
(3)    Unique ID (National ID) of the recipient of this fund: This field is required to detect duplicate/fake transactions as explained in the description later in this subsection.
(4)    Time Period of the purpose: This field is also one of the fields that are used to detect duplicate transactions.
(5)    The description of the purpose.
(6)    The valid range of amount.
(7)    Whether this row is currently valid or invalid. If invalid, then signatures of at least N NGOs and M donor agencies that decided to invalidate it.
(8)    Signatures of at least N NGOs, and at least M donor agencies.
(9)    Other related fields.

Whenever a new transaction is proposed on the network, the nodes confirm if it is not a duplicated transaction, as well as that the purpose and amount are according to the rules in the table of purposes. The duplicate transaction is defined as two different transactions, that allocate funds, for the same purpose, during the same time period, to the same recipient. Hence, the triplet (purpose ID, recipient ID, time period), cannot be repeated in the ledger.

Note that purpose ID and recipient ID are allowed to repeat multiple times if the time periods are distinct. For instance, one fund could be given to a student to buy course books in grade 7. Now consider another transaction that is giving funds for the same student, and for the same purpose of buying course books, but in a different grade, say grade 8.

The table of purposes should be allowed to change over time. One reason for this is that a change in tax and other laws can change the amount required for certain purposes. This can lead the nodes to invalidate a previous row and insert an updated row in the table. Another reason for adding a row in the table is that a new purpose for a transaction could have become possible, because of expansion in the business of the NGOs and/or donor agencies.

This table is essential in defining the policy of all connected nodes; therefore, an update would be allowed to the table only if it is signed by not just two, but a significant number of NGOs as well as donor agencies. These signatures are included in the description of the fields above. These signatures imply, that N NGOs and M donor agencies have put their reputation on the line, in case something is found faulty in this new proposed row. The values of N and M would be hardcoded in the software.

### 4.4. Adapting to the Case When Transaction Data Is Not Public

This article has so far assumed that there is complete transparency of information. That is, any node that can see the ledger can view the details of all the transactions, as well as the identities of the participating entities in any given transaction. Such a degree of transparency might not be preferred in certain situations.

To keep the details of the transaction private, it is not the transaction but the encrypted transaction that is placed on the ledger. The transaction is encrypted using the private key of one of the participating members. Let us call this member, Member A. This Member A

has to declare that it is the party whose private key is being used. Member A does this, by providing a signature of the encrypted transaction (see item 2 in Figure 5).

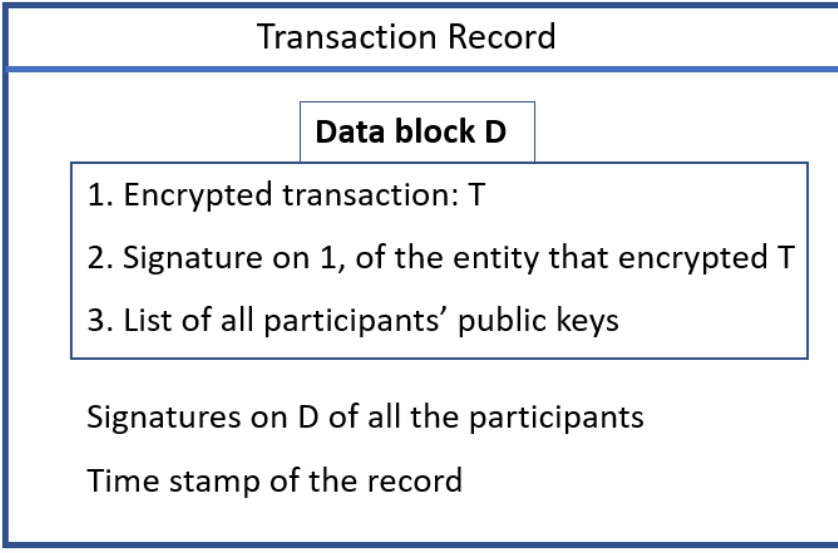

**Figure 5.** Transaction record that hides the private details of the transaction.

The record also includes the public keys of the participating parties: The NGO and the donor agencies. This block of data D: the encrypted transaction, the signature by the encrypting entity, as well as the public keys of all the participants, is then signed by the same parties that are listed. This is shown in Figure 5 as the second last item: "Signature on D of all the participants". If this list of signatures is different from the list of participants, then it is an invalid set of data, and will not be placed on the ledger.

As explained in Section 4.1.1 (for the case when the data are transparent), here it is also not possible for a fraudulent entity E to add its signature. If E is not a participant in the original transaction and tries to add its signature, then that would make the transaction record inconsistent. This is because if E adds its public key to the list of participants, it would invalidate the signatures of the original parties, as their signature was on a data set that did not include E's public key. On the other hand, if E only added its signature without adding its public key, then the record is invalid, as E's public key is not included in the list of participants, even though E's signature is included.

The parties involved in the transaction are the ones that can verify that the details of the transaction are consistent. For instance, they can confirm that the purpose of the transaction and the amount allocated is consistent according to the table of purposes. Note that the verification procedure is similar, although now as the transaction is encrypted on the ledger, only the NGOs and donors involved in the transaction can look into the transaction and verify. Furthermore, if there is ever a need to look into the details of the transaction, everyone knows who to contact, as the identification of the (NGO or donor) that encrypted the transaction, is publicly known.

## 5. Conclusions

In this research, a blockchain-based solution was proposed that would increase transparency and trust between NGOs and donor agencies in third-world countries. Some of the interesting objects and data structures that were used are: (i) an invalidation record that can be inserted into the blockchain to invalidate a previous transaction; (ii) a time-keeping table that each node maintains, to allow an entity, whether NGO or donor, to insert its records only if has participated as an active member in the system; and (iii) a table of purposes, that lists a valid range of a sum of money, for a given purpose, and is also used to detect duplicate/fake claims of two distinct funds.

This article also includes a description of some of the distributed algorithms employed including: (i) how nodes calculate whether the signatories of a transaction have spent enough time to be allowed to insert records into the ledger; (ii) how transaction records and invalidation records are verified for correctness and inserted into the ledger by consensus; and (iii) how the nodes agree on when a set of outstanding records can be made into the next block of the blockchain.

It is hoped that this detailed solution given in this article leads to an implementation that has the impact that is being envisaged: significantly improving the amount of trust and transparency in the transactions among the NGOs and donor agencies of a third world country. The implementation of the model is under process and the results will be published shortly. Furthermore, the Raft consensus algorithm is being implemented.

**Author Contributions:** Conceptualization, E.R., M.A.K., T.R.S., N.T., M.A.A. and T.M.G.; methodology, E.R., M.A.K., T.R.S., N.T., M.A.A. and T.M.G.; project administration, E.R. and M.A.K.; supervision, T.R.S.; validation, M.A.K. and T.R.S.; visualization, E.R., M.A.K. and T.R.S.; writing—original draft, E.R., M.A.K., N.T., M.A.A. and T.M.G.; writing—review and editing, E.R., T.R.S., N.T., M.A.A. and T.M.G. funding acquisition, not applicable; All authors have read and agreed to the published version of the manuscript.

**Funding:** This research received no external funding.

**Institutional Review Board Statement:** Not applicable.

**Informed Consent Statement:** Not applicable.

**Data Availability Statement:** Data Sharing not applicable. No new data were created or analyzed in this study. Data sharing is not applicable to this article.

**Conflicts of Interest:** The authors declare no conflict of interest.

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
