# Peer review of "Using Blockchain to Ensure Trust between Donor Agencies and NGOs in Under-Developed Countries"

_computers, doi:10.3390/computers10080098_

Round 1
Reviewer 1 Report
The article presents a blockchain-based solution to a well-defined problem. The problem consists of duplication in beneficiaries of NGOs of underdeveloped countries, implying that misuse of funds is present. The number of actual beneficiaries is smaller than recorded beneficiaries; therefore, the rule of a maximum of one NGO per beneficiary is broken. The paper offers a blockchain-based solution to ensure trust between donor agencies worldwide and NGOs of underdeveloped countries. The list of National IDs, along with other keys, would be available publicly on a blockchain. The distributed software would ensure that the same set of keys are not entered twice in this blockchain. The details of the fund provided to the student would also be available on the blockchain and encrypted and digitally signed by the NGOs. If a record inserted into this blockchain is considered fake, the system provides a way to cancel that record.
In this article, the authors are offering a solution for the lack of trust the donor agencies began to have in the NGOs of under-developed countries. Thereby, using a blockchain-based solution, they will increase the trust between the donor agencies and NGOs since the donor agencies will gain access through this solution to some beneficiaries keys that will be unable to be duplicated, making the NGOs unable to misuse the received funds.
The article captures your attention if you are interested in blockchain-based solutions, proposing a solution to ensure trust between donor agencies and NGOs of under-developed countries. I think the proposed solution could have a significant impact on the relationship between donor agencies and NGOs while at the same time being easy to implement. The article contains sufficient diagrams to solidify the information presented and also a good amount of references. The paper's layout is not well centered, in my opinion, and the text overlaps in the references section.
The references section must be modified so that the text does not overlap. I would also suggest aligning the text properly.
In my opinion, the article is well structured, presents some figures and tables about the architecture and research done. Also, the article is well written, in a scientific paper style and it’s readable for both novice and advanced users.
More details about the blockchain technology should be provided and performance results from a simulation/practical implementation should be provided, or at least envisioned as future work in the conclusions.
Also, the references should be updated, some of them are very old (e.g. 1987, 1995), and more related work regarding distributed ledger technologies for NGOs and crowdfunding should be provided, for example:
- Nguyen, Loan TQ, et al. "The role of blockchain technology-based social crowdfunding in advancing social value creation." Technological Forecasting and Social Change 170 (2021): 120898.
- Nadrag, Carmen, et al. "Comparative analysis of distributed ledger technologies." 2018 Global Wireless Summit (GWS). IEEE, 2018.
- DUBEY, RAMESHWAR, et al. "Blockchain for humanitarian supply chain." Supply Chain 4.0: Improving Supply Chains with Analytics and Industry 4.0 Technologies (2021): 61.
Author Response
Response to Reviewer #1
Thank you very much for your positive feedback. We appreciate your comments. Here are the answers to your comments which required our response.
1) The references section must be modified so that the text does not overlap. I would also suggest aligning the text properly.
Response: The reference section modified as per Journal provided template.
2) More details about the blockchain technology should be provided and performance results from a simulation/practical implementation should be provided, or at least envisioned as future work in the conclusions.
Response: The implementation of our model is in progress. The results will be published as a separate research paper. This fact has been included in the conclusion section of the paper.
3) Also, the references should be updated, some of them are very old (e.g. 1987, 1995), and more related work regarding distributed ledger technologies for NGOs and crowdfunding should be provided, for example:
- Nguyen, Loan TQ, et al. "The role of blockchain technology-based social crowdfunding in advancing social value creation." Technological Forecasting and Social Change 170 (2021): 120898.
- Nadrag, Carmen, et al. "Comparative analysis of distributed ledger technologies." 2018 Global Wireless Summit (GWS). IEEE, 2018.
- DUBEY, RAMESHWAR, et al. "Blockchain for humanitarian supply chain." Supply Chain 4.0: Improving Supply Chains with Analytics and Industry 4.0 Technologies (2021): 61.
Response: All the suggested references are included in the related work section.
Reviewer 2 Report
First of all, the paper is poorly written and requires a lot of improvement.
Most of the figures are poorly drawn. Please increase the quality of the figures.
The paper's novelty is low, and authors just propose blockchain for NGOs, which is not enough contribution.
Hundreds of paper available in similar directions where authors developed such framework using blockchain. Without any actual scientific contribution and without any real implementation, such papers do not provide any scientific contribution.
Paper requires more and more technical details including implementation code. e.g. Which consensus they are using, how they are mining blocks, how they improved throughput etc.
Author Response
Response to Reviewer #2
Thank you very much for your positive feedback. We appreciate your comments. Here are the answers to your comments which required our response.
1) Most of the figures are poorly drawn. Please increase the quality of the figures.
Response: All figures are modified to high quality as per the comments of reviewer 2.
2) The paper's novelty is low, and authors just propose blockchain for NGOs, which is not enough contribution.
Response: In addition to proposing a blockchain solution for NGOs, this research has shown how to solve a very specific duplication fraud in the educational funds of a third-world country. A second point is, that in section 3.5, a novel penalty mechanism is proposed to encourage the members to stay connected to the system and contribute to it. Regarding the novelty of the proposed work, reviewer 1 comments are very much encouraging for the authors.
3) Hundreds of paper available in similar directions where authors developed such framework using blockchain. Without any actual scientific contribution and without any real implementation, such papers do not provide any scientific contribution.
Response: This paper provides the solution to build trust among the donors and NGOs of under-developing countries. The main novelty of the paper is an application of known algorithms and tools, such as blockchain, public key, private key, and then combine them to solve a new specific problem. The paper has discussed the model in detail so that it will be enough to solve the problem. Although the implementation will be done in the future, and it could be a long process, but it would not offer any unsurmountable technical difficulties. In this regard, the review 1 comments are very much encouraging for the authors.
4) Paper requires more and more technical details including implementation code. e.g. Which consensus they are using, how they are mining blocks, how they improved throughput etc.
Response: The implementation of our model is in progress by using the Raft consensus algorithm which is equivalent to Paxos in performance. This fact has been included in the conclusion section of the paper.
Although not every detail of the mining algorithm being used has been provided (several pre-existing mining techniques can be implemented), however important variables to consider when mining the blocks, and important departures from the usual techniques has been described in detail in the following sections:
4.1: The different kinds of records (there is more than one type) in the ledger.
4.3.1 and 4.3.2: Addition of block into the ledger: Under what circumstances would a block not be added to the ledger. This includes duplication (or fraudulent additions), as well as insufficient online time spent by the node (if a particular NGO/Donor only joins the network to give its block, while not contributing to the processing in any way, then a certain penalty is levied, and the node has to wait before its block is added to the ledger).
4.3.3: The frequency with which the blocks are to be added to the ledger.
4.4: How to detect duplicates in the ledger and hence avoid inserting them
Round 2
Reviewer 1 Report
All comments have been solved
Author Response
Thank you very much for your positive feedback. We appreciate your comments.
Reviewer 2 Report
Great, you can also use draw.io next time for good figures. You will get good quality.
You mentioned that you are using Raft consensus. But you imported many rules and regulations to import a new block that completely changes the whole consensus, and therefore it is no more Raft consensus. Now it is important to see how your consensus is working practically. If not whole implementation, only that part of the consensus algorithm used to implement these conditions may be interesting for readers. Please include only those conditions in the form of code or algorithm. No full implementation.
Regarding novelty, you mentioned that “The main novelty of the paper is an application of known algorithms and tools, such as blockchain, public key, private key, and then combine them to solve a new specific problem.” But you can see several blogs who already proposed it.
If you google, you will see more results. All blockchain already uses public and private keys, so what is new here? You said Reviewer 1 encouraged novelty, but in the review process, we are supposed to provide independent reviews, and then the editor decides what to do.
At one point, you mentioned that implementation will be done in future and in other statements you mentioned that implementation is in process by Raft consensus. Please read how Raft consensus work, in this way you will know why Raft cannot solve your problem.
Several wrong statements in the paper clearly shows lack of blockchain basic knowledge. For example: you mentioned the block contains a fixed number of transactions. Where you read this ?
You put certain conditions that donor will sign or NGO will sign, but as per my knowledge, sender and receiver in any blockchain network already sign and its basic property.
How you implement this penalty mechanism for a distributed network? Its serious problem of all blockchain. Maybe you can write a short algorithm and how you imposed it in Raft consensus.
Check all three properties to add a new block in the blockchain. If NGO and donor both sign the transaction, specify fund and transaction is not duplicate, then it can be added to block. Is this only condition? How will some donors know that you actually used funds and not lying?
There are several questions that reader can arise technically and theoretically. I would suggest reading more good papers, properly implementing your algorithm, and then submitting the same paper again. Once you have your code, all the answers will be automatically available to reviewers or readers.
Last but not least, don’t write a technical story but do some actual work to help others. Your paper does not contain any algorithm, any code, and mathematics, any graph. It indicates that it’s not a scientific/technical paper. However, such a paper might be good for information management journals after improving the writing qualities.
Author Response
Thank you very much for your positive feedback. We appreciate your comments. Here are the answers to your comments which required our response.
Great, you can also use draw.io next time for good figures. You will get good quality.
Thanks
You mentioned that you are using Raft consensus. But you imported many rules and regulations to import a new block that completely changes the whole consensus, and therefore it is no more Raft consensus. Now it is important to see how your consensus is working practically. If not whole implementation, only that part of the consensus algorithm used to implement these conditions may be interesting for readers. Please include only those conditions in the form of code or algorithm. No full implementation.
Dear reviewer, thank you for the constructive remarks, we appreciate it, and we incorporated several changes.
Two whole new sections (3.4 and 4.3.2.2) have been added giving psuedo-code, and line-by-line explanation of a couple of algorithms in the proposed article. In one of those, the probabilistic analysis of the algorithm is also performed.
These two algorithms, now explained in detail are:
(1) How a node joins the network (depending on whether the node is trusted)
(2) Levying a penalty on those nodes that do not spend enough time connected to the distributed system, by not accepting their records into the block.
The second one is more interesting and leads to an analysis of the probability of rejecting a record when it should have been accepted.
The adaptation of the Raft algorithm for consensus would be performed in the future.
Regarding novelty, you mentioned that “The main novelty of the paper is an application of known algorithms and tools, such as blockchain, public key, private key, and then combine them to solve a new specific problem.” But you can see several blogs who already proposed it.
We agree that there have been other proposed blockchain based applications for NGOs, however, there are differences (for instance compared to the website link above). Firstly, the current article proposes cancellation of the records, secondly it proposes a new penalty mechanism, and thirdly there is a finite list of ‘purposes’ in a table, with each purpose having a designated range of funds that can be allotted for it.
If you google, you will see more results. All blockchain already uses public and private keys, so what is new here?
We do agree with the reviewer. As far as what is new here is already replied in the above comment.
You said Reviewer 1 encouraged novelty, but in the review process, we are supposed to provide independent reviews, and then the editor decides what to do.
Understood and agreed. It probably came off in the wrong way. It was not our intention to make a comparison, and we apologize if it seems that we are doing so.
At one point, you mentioned that implementation will be done in future and in other statements you mentioned that implementation is in process by Raft consensus. Please read how Raft consensus work, in this way you will know why Raft cannot solve your problem.
Raft was mentioned only in the future work. We do believe that Raft can be adapted for at least certain parts of our application. We would only know for sure when the implementation has been completed.
Several wrong statements in the paper clearly shows lack of blockchain basic knowledge. For example: you mentioned the block contains a fixed number of transactions. Where you read this ?
Thank you for identifying our mistake. The said statement has been fixed as :
The block contains transactions that store information about some important activity, such as tracking goods or assets, sensitive medical information, or critical information generated by machines using IoT [6].
You put certain conditions that donor will sign or NGO will sign, but as per my knowledge, sender and receiver in any blockchain network already sign and its basic property.
Agreed. They were mentioned for completion and emphasis.
How you implement this penalty mechanism for a distributed network? Its serious problem of all blockchain. Maybe you can write a short algorithm and how you imposed it in Raft consensus.
The pseudo-code (section 4.3.2.2) and probabilistic analysis of the penalty mechanism is now included in the article. The algorithm as it stands does not depend on raft.
Check all three properties to add a new block in the blockchain. If NGO and donor both sign the transaction, specify fund and transaction is not duplicate, then it can be added to block. Is this only condition? How will some donors know that you actually used funds and not lying?
In the real world, it is extremely difficult to find if someone is lying or not. So, the same problem can be viewed in online systems. Such types of cases can be easily found on Google (fake account cases). If all required fields are provided then the transaction will be accepted by our application. Certain types of lies can be detected by our application but it is impossible to detect all kinds of lies.
There are several questions that reader can arise technically and theoretically. I would suggest reading more good papers, properly implementing your algorithm, and then submitting the same paper again. Once you have your code, all the answers will be automatically available to reviewers or readers.
We will indeed read more good papers, and implement. Thank you for your constructive advice.
Also, to increase readability and clarity:
(1) A few of those paragraphs have been rewritten, where we thought clarity could have been improved
(2) The algorithm of penalty mechanism had included decaying weights. This decay factor has now been taken out, as the algorithm can stand without the decay factor. This simplification should increase readability. It also made a straightforward mathematical analysis of the algorithm a possibility.
Last but not least, don’t write a technical story but do some actual work to help others. Your paper does not contain any algorithm, any code, and mathematics, any graph. It indicates that it’s not a scientific/technical paper. However, such a paper might be good for information management journals after improving the writing qualities.
Pseudo-code (algorithms), and a probabilistic analysis of one of them, is now included in the article.
